# Clinical Outcomes and Cost Implications of a Community Psychosocial Rehabilitation Service for Severe and Persistent Mental Illness in Nova Scotia, Canada

**DOI:** 10.3390/healthcare12181904

**Published:** 2024-09-23

**Authors:** Mahmoud A. Awara, Joshua T. Green

**Affiliations:** 1The Royal College of Psychiatrists, 21 Prescot St., London E1 8BB, UK; 2Faculty of Medicine, Dalhousie University, 1459 Oxford Street, Halifax, NS B3H 4R2, Canada; josh.green@nshealth.ca; 3Department of Psychiatry, Nova Scotia Health, QEII Health Sciences Centre, Veterans Memorial Lane, 8th Floor, Abbie J. Lane Memorial Building, Halifax, NS B3H 2E2, Canada; 4The College of Physicians and Surgeons of Nova Scotia, Suite 400, 175 Western Parkway, Bedford, NS B4B 0V1, Canada; 5Nova Scotia Health Authority, 1276 South Park Street, Halifax NS B3H 2Y9, Canada

**Keywords:** clinical effectiveness, psychosocial rehabilitation

## Abstract

**Introduction:** Individuals with severe and persistent mental illness (SPMI) present distinct challenges in mental healthcare due to the chronic and complex nature of their conditions. This study was conducted to assess the clinical efficacy and potential cost-effectiveness of a multidisciplinary community-based psychosocial rehabilitation team serving individuals with SPMI in Nova Scotia, Canada. **Method**: This study was conducted to evaluate the effects of a community-based psychosocial rehabilitation program on individuals with severe and persistent mental illness (SPMI) in Nova Scotia, Canada. This research focused on clinical outcomes and potential cost savings following a one-year intervention, offering critical insights into the benefits of community-based care for this population. A cohort of 137 clients accepted into the community rehabilitation service (Connections Dartmouth) between September 2016 and September 2020 was analyzed. Each participant received one year of community rehabilitation intervention. Using data from the Canadian Medical Service Insurance (MSI) billing system, this research compared the use of inpatient services and Emergency Department visits in the year prior to and the year following the intervention. The findings provide valuable evidence on the role of community rehabilitation in reducing healthcare utilization for individuals with SPMI. **Results:** The results demonstrated a statistically significant reduction in mean admission rates and length of inpatient admissions in the year following rehabilitation compared to the pre-rehabilitation year. A substantial percentage of patients experienced no inpatient admissions (88% vs. 60%) or Emergency Department visits (82% vs. 67%) in the post-rehabilitation year, compared to the pre-rehabilitation year. There was a significant reduction in inpatient days by 90%, translating into substantial cost savings. The findings highlight the potential economic benefits of community rehabilitation for people with SPMI. **Conclusions:** This uncontrolled study suggests that community rehabilitation is associated with positive clinical outcomes for individuals with SPMI in terms of reduced inpatient service use and associated costs. Further research, including controlled studies and cost-effectiveness studies, into the community psychosocial rehabilitation services in the Canadian setting is needed.

## 1. Introduction

People with severe and persistent mental illness (SPMI) are relatively low in number but consume a high proportion of mental healthcare resources due to the complexity and longer-term nature of their mental health problems. The term ‘SPMI’ was introduced by the National Institute of Mental Health (NIMH) in 1987 to replace “chronic mental illness” (CMI), in part to address the negative associations and therapeutic pessimism associated with the word ‘chronic’ [1]. The NIMH definition of SPMI comprises three dimensions: diagnosis, disability, and duration. People with SPMI usually have a diagnosis of schizophrenia, schizoaffective disorder, bipolar affective disorder, or personality disorder, with functional limitations in managing usual life activities, and the condition is likely to be persistent through the adult lifespan. The majority are unemployed or require supportive work environments, most receive welfare benefits and require assistance to manage everyday activities, and many struggle with interpersonal skills and relationships as a result of their illness [1].

In Canada, individuals with SPMI face numerous challenges, including limited access to specialized community rehabilitation services, high rates of hospitalization, and inadequate post-discharge follow-up, all of which contribute to an overreliance on inpatient care. Research has shown that inadequate community-based services lead to higher costs and worse long-term outcomes for patients with SPMI (Boydell et al. 2003; Sealy and Whitehead 2004). This study addresses these gaps by evaluating the impact of a community-based rehabilitation team in Nova Scotia [2,3].

The UK’s National Institute for Health and Care Excellence (NICE) recently published clinical guidance on the mental health rehabilitation services and interventions that should be available to adults with SPMI (NICE, 2020). This guidance recommends that local rehabilitation services should comprise both inpatient rehabilitation units and community-based supported accommodation services that receive specialist clinical input from multidisciplinary community rehabilitation teams. These services should be embedded within the wider local mental health system and organized into a defined rehabilitation care pathway that supports people to progress in their recovery, with the aim of stabilizing symptoms and enabling people to gain the confidence and skills to live successfully in the community and achieve optimum autonomy. All rehabilitation services should adopt a recovery-oriented approach that is, by definition, collaborative and individualized [4] (https://www.nice.org.uk/guidance/ng181 (accessed on 21 September 2023)).

Growing evidence from the United Kingdom highlights the clinical effectiveness of mental health rehabilitation services. Approximately two-thirds of individuals receiving support from these services transition to successful community living within 18 months of admission to a National Health Service (NHS) inpatient rehabilitation unit, with two-thirds maintaining this progress over five years without requiring further hospitalizations [5]. Additionally, around 10% achieve independent living within this timeframe [5]. Patients engaged with rehabilitation services are eight times more likely to achieve and sustain community living compared to those receiving support from generic community mental health services [5,6]. In the absence of specialized mental health rehabilitation, individuals with severe and complex mental health disorders face a significantly higher risk of self-neglect, exploitation, and long-term institutionalization [6].

A recent systematic review (Dalton-Locke et al. 2021) of the evidence for the effectiveness of inpatient and community-based mental health rehabilitation services included 65 studies conducted across 14 countries. Inpatient rehabilitation was associated with a reduction in subsequent acute inpatient service use. However, once in the community, only around half of the people with SPMI graduated from higher to lower levels of supported accommodation within the expected timeframes [7]. One of the included studies by Bunyan et al. (2016) also showed reductions in inpatient service costs associated with inpatient rehabilitation due to the reduced inpatient service use following the rehabilitation admission [8]. A further study investigated the effectiveness of a specialist inpatient facility (the National Psychosis Unit) that provides evidence-based, personalized, multidisciplinary interventions for patients with complex psychosis in London, UK. This two-year ‘mirror image’ study reported that the service was associated with reduced inpatient service use [9].

Dalton-Locke’s systematic review (2021) [7] has also identified eight studies that evaluated community rehabilitation teams. These included teams delivering manualized models of psychosocial support (such as the Illness Management and Recovery (IMR) program, which primarily involves psychoeducation and personal recovery promotion delivered through group sessions over several months) and teams providing case management and other less-defined approaches. The findings were mixed, likely due partly to the interventions’ heterogeneity and differences in the target client group and the context within which the studies were conducted (for example, whether the service was part of a broader mental health system) [10,11,12,13,14,15,16,17].

Assertive Community Treatment (ACT) is a well-established model of community-based care for individuals with SPMI, emphasizing comprehensive, integrated services, delivered in clients’ communities (Marshall and Lockwood 2000). Research consistently demonstrates its effectiveness in reducing hospitalization rates and improving functional outcomes (Bond et al. 2001). Our study builds on these principles by evaluating a similar community rehabilitation approach within the Canadian healthcare context [18,19].

The clinical outcomes of psychosocial rehabilitation services have not been fully studied in Canada. This study was conducted to address this by evaluating one community rehabilitation service based in Dartmouth, a city in Nova Scotia, Canada, known as ‘Connections Dartmouth’. The primary hypothesis was that care from the community rehabilitation service would be associated with a reduction in inpatient service utilization, as evidenced by decreases in admissions, length of stay, and emergency services presentations.

## 2. Method

### 2.1. Study Design

A comparative study examining inpatient service utilization and emergency room visits in the 12 months before enrollment in the community rehabilitation program and the 12 months following a full year of engagement with the community rehabilitation team.

### 2.2. Setting

Connections Dartmouth is a community mental health rehabilitation team with an average caseload of 200 clients with SPMI. It is part of the Recovery and Integration service portfolio of the Nova Scotia Health “Mental Health and Addictions Program”, which provides community-based rehabilitation services for a total population of 420,000. It is one of three recovery-oriented local services that support adults with SPMI. The team receives referrals from several services, e.g., forensic services, acute inpatient services, other community mental health teams, early psychosis services, primary care, and child and adolescent services.

The service is staffed by a multidisciplinary team of psychiatrists, social workers, occupational therapists, and registered nurses with individual case manager caseloads between 12 and 20. Recreation therapists, an occupational therapy assistant, a developmental worker, and a secretary support the team.

Members of the team provide services on-site and outreach to community settings. Many clients experience a high degree of complexity in addition to mental illness, including trauma history, behavioural disorders, substance misuse, cognitive impairment, and significant psychosocial challenges, such as homelessness and poverty. The team works with each client towards personalized and self-identified psychosocial goals. These vary widely and comprise addressing direct mental health issues through symptom reduction as well as working on environmental aspects that impact psychological well-being. The latter includes but is not limited to improving clients’ housing situation, employment, or education opportunities to improve skills and prospects, reducing social isolation and building relationships within a community that provides friendship, empathy, support, and hope. The aim is for the client to be an active partner in their own recovery plan while being supported by the team, with the goal of living a meaningful life in their community.

The service offers facilitated group programs to build skills (e.g., cooking/baking, running, yoga, mindfulness, social skills). Clients can also join community-based work skills programs (e.g., carpentry, rubbish collection from hospitals, cleaning, etc.). The team also works closely with local voluntary sector organizations that provide additional support and employment opportunities.

### 2.3. Sample

The study examined a cohort of 137 clients accepted by the community rehabilitation service (Connections Dartmouth) between September 2016 and September 2020 and received at least one year of community rehabilitation intervention.

### 2.4. Data

All participants’ Medical Record Numbers (MRNs) were used to analyze service utilization for one year before joining the service (pre-treatment) and one year following one full year of treatment (post-treatment).

Acute inpatient admissions and length of stay were recorded from various mental health units within the Central Zone of the Nova Scotia Health Authority. ED visits were limited to mental health and addiction-related cases at regional hospitals and were gathered from the Emergency Department Information System (EDIS). The data, including psychiatric diagnoses, admissions rate, and length of stay, were retrieved from the Discharge Abstract Database (DAD).

The costs of the service utilization were estimated during the pre-and post-rehabilitation years using the Canadian billing system of the Medical Service Insurance (MSI) in Nova Scotia, Canada: using the Canadian Dollar (CAD) 1450 per patient day for inpatient acute mental health care and CAD 359 for an ER visit or outpatient community rehabilitation visit.

### 2.5. Data Analysis

Descriptive statistics were calculated, including mean, median, standard deviations, and 95% confidence intervals. Due to the non-normal distribution of the data, comparisons of inpatient service use and emergency room presentations in the year before being taken on by the rehabilitation service and the year after were conducted using Wilcoxon matched-pairs signed-ranks tests. Statistical analyses were performed at a 95% confidence interval (α = 0.05) using SAS JMP version 15.0.

## 3. Results

Most of the team’s clients were male, with a mean age of 44 years. The primary diagnosis was schizophrenia (see Table 1 and Table 2).

As illustrated in Table 3, the mean admission rate and mean LOS were statistically significantly lower in the post-rehabilitation year than in the pre-rehabilitation year. In addition, 88% of patients in the post-rehabilitation year had no inpatient admissions, compared to 60% in the pre-rehabilitation year. Similarly, 82% of patients in the post-rehabilitation year had no visits to the ER for mental health, compared to 67% in the pre-rehabilitation period.

The number of mental health related visits to the Emergency Department has dropped from 123 before treatment to 47 after treatment. When comparing the mean of visits in the year before (0.89) and after (0.34) being taken on by the community rehabilitation team, the Wilcoxon matched-pairs signed-rank test *p*-value indicates a significant reduction (*p* = 0.005).

### Costs

The reduction in post-rehabilitation hospital stays, reflecting a decrease of 6785 patient days at a daily cost savings of CAD 1450.00 (Canadian Dollar), results in a cumulative estimated saving of—CAD 9,838,250.00. Additionally, the evaluation of MHA-related ED visits, employing an average cost per visit of CAD 359.00 and considering a reduction of 76 visits, demonstrates a net cost saving of—CAD 27,284.00.

In the context of community rehabilitation, the cohort comprising 137 clients underwent a total of 1050 visits to a community rehabilitation clinic over a year. With an established average cost of CAD 359.00 per outpatient visit, the comprehensive estimated expenditure for all visits amounts to CAD 376,950.

The culmination of these analyses, when factoring in costs associated with community rehabilitation treatment, inpatient hospitalizations, and ED visit reduction, reveals a net saving of CAD 9,488,584. This examination underscores the fiscal advantages arising from implementing community psychosocial rehabilitation intervention within the healthcare paradigm.

## 4. Discussion

The present study makes a significant contribution to the understanding of the importance of community rehabilitation services for individuals with severe persistent mental illness (SPMI) in Canada. This study evaluated the impact of a community rehabilitation team on inpatient service use and Emergency Department (ED) visits among individuals with SPMI over one year before and one year after 12 months of engagement with the rehabilitation team. The findings revealed substantial reductions in inpatient service use and ED visits and significant associated cost savings. A notable percentage of patients experienced no inpatient admissions or ED visits following a year of rehabilitation. These findings underscore community mental health rehabilitation’s potential economic and clinical benefits for this patient group.

Dalton-Locke et al., in their systematic review evaluating the effectiveness of mental health rehabilitation services, found consistent evidence for the effectiveness of inpatient rehabilitation services, particularly in terms of subsequent reductions in inpatient service utilization, but the impact of community rehabilitation services was less clear [7]. This study provides evidence that community rehabilitation teams are clinically effective, and the strengths of this study provide greater confidence in these results, including the relatively large sample size, the focus on people with SPMI, and the use of robust (‘hard’) outcome measures, including admission rates, duration of hospital stays, and visits to Emergency Departments.

However, we also acknowledge the study’s limitations, including the absence of a control group and randomization. Nevertheless, the demonstrated service-use reductions are unlikely to be explained entirely by regression to the mean. In addition to analysis of similar services across Canada, future studies could examine outcome measures between various local mental health services, particularly services specializing in the treatment of primary psychotic illnesses (e.g., early psychosis programs), to better delineate the effectiveness of rehabilitation services in this population.

This study aligns with the resurgence of interest in mental health rehabilitation, as emphasized in recent NICE guidelines [4] highlighting the importance of these services for people with complex psychosis. The findings contribute to the evolving understanding of the need for this subspecialty for individuals with SPMI.

In the context of ongoing economic austerity and the increasing demand for efficient service delivery, assessing the cost-effectiveness of psychosocial rehabilitation interventions is paramount. Although this study does not offer a comprehensive evaluation of the service’s cost-effectiveness, the observed reductions in service utilization and related costs are promising. While detailed cost data for the community rehabilitation service were not fully reported in our analysis, the long-term benefits—particularly sustained reductions in service use—suggest the potential for these savings to offset the initial treatment costs.

Despite the scarcity of research on community psychosocial rehabilitation in Canada, this study provides valuable insights into the potential effectiveness of such services. The authors hope this research will inspire further investigations into the clinical and cost-effectiveness of community mental health rehabilitation teams.

Future research should focus on controlled studies to assess the efficacy of psychosocial rehabilitation services compared to other interventions. Long-term studies are also needed to evaluate the sustainability of the observed outcomes (Killaspy et al. 2006). Additionally, research exploring the effectiveness of various psychosocial rehabilitation models, such as Assertive Community Treatment (ACT) and the Illness Management and Recovery (IMR) program, would help clarify the most effective approaches for this population (Mueser et al. 2002) [20,21].

## 5. Conclusions

This study adds to the growing body of literature supporting the potential effectiveness of psychosocial rehabilitation services for individuals with SPMI in Canada. The positive outcomes, including significant reductions in inpatient service use and associated cost savings, underscore the importance of further research and the potential economic benefits of investing in community psychosocial rehabilitation.

## Figures and Tables

**Table 1 healthcare-12-01904-t001:** Principal diagnoses among the study population.

Principal Diagnosis	N = 137	% of Total
Schizophrenia	76	55%
Schizoaffective disorder	14	10%
Other psychotic disorder	13	9%
Drug-related or induced disorders with a comorbidity of personality disorder	13	9%
Drug-related or induced disorders	11	8%
Personality disorder	4	3%
Bipolar affective disorder	3	3%
Adjustment disorder	1	1%
Depressive disorder	1	1%
Dissociative psychosis	1	1%

**Table 2 healthcare-12-01904-t002:** Study sample demographics.

Sex	Mean Age	Median Age	N	% of Total
Female	49	52	52	38%
Male	40	37	85	62%
Employment	-	-	0	0
**All**	**44**	**44**	**137**	**100%**

**Table 3 healthcare-12-01904-t003:** Comparative analysis of inpatient service utilization before and after intervention by a community rehabilitation team.

Variable	Inpatient Mental Health Admissions		Inpatient Mental Health Days	
	12 Months before Receiving Community Rehab	12 Months Following One Year of Community Rehab	12 Months before Receiving Community Rehab	12 Months Following One Year of Community Rehab
No. of admissions/No. of days	76	26	7469 days	684 days
Mean	0.55	0.19	54.51	4.99
Median	0	0	0	0
IQR	1	0	58	0
Standard deviation of the mean	0.77	0.56	102.96	16.59
Lower 95% for the mean	0.42	0.09	37.12	2.18
Upper 95% for the mean	0.68	0.28	71.91	7.79
Wilcoxon matched-pairs signed-rank test *p*-value	<0.0001 **** (two-tailed)		<0.0001 **** (two-tailed)	

Key Insights of Table 3: Inpatient Admissions: After one year of community rehabilitation, the total number of admissions decreased significantly from 76 to 26, with a corresponding drop in the mean number of admissions per client (from 0.55 to 0.19). **** The statistical test Wilcoxon matched-pairs signed ranks test *p*-value < 0.0001) confirms this reduction is highly significant. Inpatient Days: The total number of inpatient days saw a dramatic reduction, dropping from 7469 to just 684. The mean number of inpatient days per client decreased from 54 to 4, again with a highly significant *p*-value < 0.0001.

## Data Availability

Additional data are available from the corresponding author upon reasonable request.

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
