# Peer review of "Clinical Outcomes and Cost Implications of a Community Psychosocial Rehabilitation Service for Severe and Persistent Mental Illness in Nova Scotia, Canada"

_healthcare, 2024, doi:10.3390/healthcare12181904_

Round 1

Reviewer 1 Report

Comments and Suggestions for Authors

The work presented is of enormous interest and relevance.  For more than 15 years I worked in the field of psychosocial rehabilitation of people with mental illness and such serious and rigorous studies are to be welcomed. They also focus on economic aspects (which are fundamental to explain many issues related to rehabilitation).

Below are a number of aspects that could improve the wonderful manuscript:

- Although the introduction mentions the term SPMI and its impact, it might be useful for readers to have a more complete description of the problem in the Canadian context. In addition, a more detailed discussion of the need for this study in Canada would greatly strengthen the rationale for the research.

- Similarly, it would be interesting to incorporate a table with a (demographic) description of the study participants. Right now, the authors only incorporate diagnostic data when other aspects (age, social class, etc) could be of enormous relevance.

- Conclude with clear recommendations for future research: end with clear and specific recommendations for future research, including controlled studies, long-term analyses and exploration of different psychosocial intervention modalities.

Author Response

Thank you very much for your positive feedback and your valuable suggestions, which have significantly enhanced our manuscript. We greatly appreciate your insights, particularly given your extensive experience in the field of psychosocial rehabilitation.

  • Clarification on the Canadian Context of SPMI: We agree that providing more context on the challenges of severe and persistent mental illness (SPMI) in Canada would strengthen the rationale for the study. To address this, we have expanded the introduction to include a more detailed discussion of the unique healthcare landscape in Canada and the pressing need for rehabilitation services for individuals with SPMI. This should provide a clearer justification for the study (second paragraph in the introduction).
  • Demographic Table for Study Participants: Your suggestion to include a demographic table is well-taken. In response, we have added a table with a demographic breakdown of the study participants, which includes variables such as age, gender, and employment status, in addition to the diagnostic data already presented. This addition should enhance the understanding of the population under study.
  • Recommendations for Future Research: We have strengthened the conclusion section by adding specific recommendations for future research, including suggestions for controlled studies, long-term follow-up analyses, and an exploration of various psychosocial rehabilitation interventions. We hope these additions provide a clearer direction for future work in the last paragraph of the discussion.

Reviewer 2 Report

Comments and Suggestions for Authors

The authors have conducted an important evaluation of a community psychosocial rehabilitation program for people with severe and persistent mental illness. This topic is highly relevant and timely. 

There are a few ways to strengthen the manuscript.

The study design (a pre-post test) has implications for which words can be used. Since the design was not an effectiveness study (such as a randomized, controlled trial), the word, Effectiveness, should not be used in the title or elsewhere in relation to the design of the study or the results. 

The readers might benefit from a review of the research on similar approach, Assertive Community Treatment, in the introduction.

Methods

The readers might benefit from additional details about the program. For example, do the multi-disciplinary team members visit the patient at home or do patients go to Dartmouth Connections to access care? Are all patients able to manage their finances? ...

Data

The data are not clear. Under 2.4 Data, were all admissions, discharges, and ER visits included in the analyses? It is written thus.  

Perhaps an identification number or names of the 137 participants were used to select cases from all the admissions, discharges and ER visits before and after implementation? 

Results

I appreciate that the first paragraph appropriately contains a description of the sample; however, additional details would be helpful including percent male, the indicator of the variability in age, were any participant employed, and so forth. 

I find the format of Table 2 makes the information difficult to read but perhaps this format was a journal requirement.

The limitations of the study should be fully reported. There are many threats to internal validity when randomization to intervention or a control group is not used, and these limitations should be acknowledged.   

Comments on the Quality of English Language

The manuscript was generally well-written; however, a few areas could be strengthened.

Abstract 

It is indicated that "The study evaluated...." which reflects anthropomorphism. People can evaluate.  People design and conduct study to evaluate something.

There is a period missing in the Results of the Abstract.

Another instance of anthropomorphism is "Dalton-Locke’s systematic review (2021) has also identified eight studies...  ."  It is Dalton Locke who conducted a systematic review and identified eight studies.

Author Response

The authors have conducted an important evaluation of a community psychosocial rehabilitation program for people with severe and persistent mental illness. This topic is highly relevant and timely. 

There are a few ways to strengthen the manuscript.

The study design (a pre-post test) has implications for which words can be used. Since the design was not an effectiveness study (such as a randomized, controlled trial), the word, Effectiveness, should not be used in the title or elsewhere in relation to the design of the study or the results. 

The readers might benefit from a review of the research on similar approach, Assertive Community Treatment, in the introduction.

Methods

The readers might benefit from additional details about the program. For example, do the multi-disciplinary team members visit the patient at home or do patients go to Dartmouth Connections to access care? Are all patients able to manage their finances? ...

Data

The data are not clear. Under 2.4 Data, were all admissions, discharges, and ER visits included in the analyses? It is written thus.  

Perhaps an identification number or names of the 137 participants were used to select cases from all the admissions, discharges and ER visits before and after implementation? 

Results

I appreciate that the first paragraph appropriately contains a description of the sample; however, additional details would be helpful including percent male, the indicator of the variability in age, were any participant employed, and so forth. 

I find the format of Table 2 makes the information difficult to read but perhaps this format was a journal requirement.

The limitations of the study should be fully reported. There are many threats to internal validity when randomization to intervention or a control group is not used, and these limitations should be acknowledged.   

 Comments on the Quality of English Language

The manuscript was generally well-written; however, a few areas could be strengthened.

Abstract 

It is indicated that "The study evaluated...." which reflects anthropomorphism. People can evaluate.  People design and conduct study to evaluate something.

There is a period missing in the Results of the Abstract.

Another instance of anthropomorphism is "Dalton-Locke’s systematic review (2021) has also identified eight studies...  ."  It is Dalton Locke who conducted a systematic review and identified eight studies.

Response to Reviewer:

We are grateful for your detailed review and insightful comments, which have been extremely helpful in refining our manuscript. We have carefully addressed each of your points as follows:

  • Terminology Related to Study Design: We acknowledge the importance of using precise terminology regarding study design. In response, we have revised the title of the manuscript to replace "Effectiveness" with  "Outcomes," reflecting the pre-post design rather than a randomized controlled trial. We appreciate your attention to this critical detail.
  • Review of Assertive Community Treatment (ACT): Thank you for pointing out the relevance of ACT to our study. We have incorporated a brief review of the research on Assertive Community Treatment in the introduction section, situating our work within the broader context of community-based mental health interventions. (the penultimate paragraph in the introduction).
  • Clarification of Program Details: Your suggestion to provide more details about the program's operations is much appreciated. We have highlighted all the support offered to clients as part of the psychosocial rehabilitation process (third paragraph under section "sitting"2.2).
  • Clarification of Data: Under data 2.4, we have revised the data section to clarify that all admissions, discharges, and emergency room visits related to mental health and addictions within the specified timeframe were included in the analysis. We have also specified that participant identifiers (Medical Record Number) were used to match pre- and post-intervention data.
  • Sample Description: In response to your suggestion, we have added additional descriptive details about the sample, including gender, age and employment status (table 2). This should provide a better understanding of the population under study. We have clarified Table (3) and added key insights to make it easier to interpret. 
  • Limitations: We agree on the study's limitations; hence, we have highlighted this section in the third paragraph of the discussion. In the last paragraph, we added suggestions that future research be designed to address these limitations. 
  • Quality of English Language: We appreciate your detailed suggestions regarding language improvements. The anthropomorphism and minor grammatical issues you highlighted have been corrected, including the rephrasing of sentences in the abstract to reflect more appropriate subject-verb relationships.
    • Abstract 

      It is indicated that "The study evaluated...." which reflects anthropomorphism. People can evaluate.  People design and conduct study to evaluate something. We corrected that e.g.,  "the study was conducted to evaluate."

    •  

      There is a period missing in the Results of the Abstract. We have clarified that point and provided details in the abstract's results section.

      Another instance of anthropomorphism is "Dalton-Locke’s systematic review (2021) has also identified eight studies...  ."  It is Dalton Locke who conducted a systematic review and identified eight studies. Yes 

Once again, we sincerely thank the reviewer for their thoughtful and constructive feedback. We believe our revisions in response to your comments have considerably strengthened the manuscript.